# Polysaccharide-Based Formulations for Healing of Skin-Related Wound Infections: Lessons from Animal Models and Clinical Trials

**DOI:** 10.3390/biom10010063

**Published:** 2019-12-30

**Authors:** Diogo Marcelo Lima Ribeiro, Alexsander Rodrigues Carvalho Júnior, Gustavo Henrique Rodrigues Vale de Macedo, Vitor Lopes Chagas, Lucas dos Santos Silva, Brenda da Silva Cutrim, Deivid Martins Santos, Bruno Luis Lima Soares, Adrielle Zagmignan, Rita de Cássia Mendonça de Miranda, Priscilla Barbosa Sales de Albuquerque, Luís Cláudio Nascimento da Silva

**Affiliations:** 1Programa de Pós-graduação, Universidade Ceuma, São Luís, Maranhão 65075–120, Brazil; diogomlr@hotmail.com (D.M.L.R.); arcarvalhojr@gmail.com (A.R.C.J.); gustavo.macedo.7@hotmail.com (G.H.R.V.d.M.); vitorlopes.ch@gmail.com (V.L.C.); ls.luscas@gmail.com (L.d.S.S.); bsilvadc@gmail.com (B.d.S.C.); deivid.martinss98@gmail.com (D.M.S.); brunoluis14@hotmail.com (B.L.L.S.); adriellyzagmignan@hotmail.com (A.Z.); rita.miranda@ceuma.br (R.d.C.M.d.M.); 2Departamento de Medicina, Universidade de Pernambuco, Garanhuns, Pernambuco 55294902, Brazil; priscillaibm@hotmail.com

**Keywords:** alginate, cellulose, chitosan, hyaluronic acid, infected wounds

## Abstract

Skin injuries constitute a gateway for pathogenic bacteria that can be either part of tissue microbiota or acquired from the environmental. These microorganisms (such as *Acinetobacter baumannii, Enterococcus faecalis,*
*Pseudomonas aeruginosa,* and *Staphylococcus aureus*) produce virulence factors that impair tissue integrity and sustain the inflammatory phase leading for establishment of chronic wounds. The high levels of antimicrobial resistance have limited the therapeutic arsenal for combatting skin infections. Thus, the treatment of non-healing chronic wounds is a huge challenge for health services worldwide, imposing great socio-economic damage to the affected individuals. This scenario has encouraged the use of natural polymers, such as polysaccharide, in order to develop new formulations (membranes, nanoparticles, hydrogels, scaffolds) to be applied in the treatment of skin infections. In this non-exhaustive review, we discuss the applications of polysaccharide-based formulations in the healing of infected wounds in animal models and clinical trials. The formulations discussed in this review were prepared using alginate, cellulose, chitosan, and hyaluronic acid. In addition to have healing actions *per se*, these polysaccharide formulations can act as transdermal drug delivery systems, controlling the release of active ingredients (such as antimicrobial and healing agents). The papers show that these polysaccharides-based formulations are efficient in controlling infection and improve the healing, even in chronic infected wounds. These data should positively impact the design of new dressings to treat skin infections.

## 1. Introduction

Wound healing is a complex biochemical and cellular process consisting of sequential and overlapping phases (hemostasis, inflammatory, proliferative, and remodeling stages) that occur in response to the physical disruption of tissue [1,2]. Skin injuries constitute a gateway for pathogenic bacteria that can be either part of tissue microbiota or acquired from the environment. These microorganisms produce virulence factors that impair tissue integrity and sustain the inflammatory phase leading for development of wounds [3,4,5]. Other factors that contribute to chronicity of lesions include the following comorbidities: neoplasias, metabolic disorders, vascular pathologies, and immunodeficiencies [6].

Several pathogens can be involved in skin infections; among them, some bacteria genus are highlighted such as *Acinetobacter, Enterococcus, Escherichia*, *Klebsiella*, *Pseudomonas*, *Staphylococcus,* and *Streptococcus* [3,7]. Chronic wounds are frequently infected by more than one species, resulting in increasing damage to host tissue [3]. These infections are usually associated with biofilm formation, an environment where microorganisms interact to produce extracellular substances that contribute to the phenomena of drug resistance and host immune evasion [8]. In addition, the biofilm is a favorable niche for dissemination of genes related to antimicrobial resistance and virulence determinants [9,10].

The treatment of non-healing chronic wounds is a huge challenge for the health services worldwide, imposing great socio-economic damages to the affected individuals [11,12]. A wide range of dressings are used to treat severe wounds. In general, the dressing should have some characteristics that are classified as fundamental or secondary requirements (Table 1). Examples of the fundamental properties include mechanical protection and cytocompatibility. The ideal dressing should have high absorption capacity, be comfortable, allow the visualization of the wound, and avoid pain in the removal. In addition, it should not provoke allergic reaction [13,14,15,16,17,18].

Traditional topical dressings (such as gauze) usually offer good absorption, but also provide desiccation of the wound and can provoke skin damage when removed. These disadvantages led to the development of new dressings with a broader spectrum of properties [20]. Different polymeric materials (synthetic or natural) are used for the development of dressings (foams, hydrogels, hydrocolloids, films, membranes), and each type has advantages and disadvantages making them suitable for the treatment of specific types of wound (Table 2).

Furthermore, the treatment of skin infections requires the incorporation of antimicrobial agents such as silver and other marketed antibiotics [20]. In some situations, topical antimicrobials are administrated in combination with systemic antibiotics [21]. However, the high levels of antimicrobial resistance have limited the therapeutic arsenal to combat skin infections [20,22]. In this sense, new bioactive compounds (with antimicrobial, healing, and/or immunomodulatory actions) are explored for the treatment of chronic and infected wounds [20,23,24].

Importantly, no existing product meets all the requirements of an ideal dressing. This scenario has encouraged the use of natural polymers, such as carbohydrates, in order to develop new formulations to be applied in the treatment of skin infections [23,32,33,34,35,36]. In this descriptive review, we present the application of polysaccharide-based formulations in in vivo experimental models of skin wounds infections induced by bacteria. The manuscripts were obtained in PubMed and were selected papers with formulations prepared with alginate, cellulose, chitosan, and hyaluronic acid (structural representations are shown in Figure 1).

## 2. Polysaccharides and Development of Healing Agents

Polysaccharides are stereoregular polymers of monosaccharides extracted from plants, algae, animals, fungi, or obtained via fermentation [37,38,39]; these complex carbohydrates are useful in many biotechnological applications, for example as dietary fiber, texture modifiers, gelling agents, thickeners, emulsifiers, stabilizers, coating agents, and packaging films [40,41,42]. They are considered unique raw materials due to their inexpensiveness and great availability, for example plant cellulose and chitosan [37,38].

In addition, it is possible to point at several biological and chemical properties that support their application as healing agents, including non-toxicity, biocompatibility, biodegradability, poly-functionality, high chemical reactivity, chirality, chelation, and adsorption capacity [41,43]. The excellent adsorption behavior of polysaccharides is related to four properties: (1) Their high hydrophilicity; (2) the presence of a vast number of functional groups (acetamido, primary amino, and/or hydroxyl groups); (3) the high chemical reactivity of these functional groups; and (4) the flexible structure of the polymer chain [44,45].

Natural gums are polysaccharides composed of multiple sugar units that crosslink to create large molecules with heterogeneous composition. Upon hydrolysis, they yield simple monosaccharides, such as arabinose, galactose, glucose, mannose, xylose or uronic acids [46,47]. Gums are abundant in nature and commonly found in many higher plants; in addition, they are frequently produced as a protection mechanism following plant injury [48,49]. Besides gums, the polysaccharides also include mucilages. Although they share a natural polymeric source, gums and mucilages have certain differences: (i) gums readily dissolve in water, while mucilages form viscous masses; (ii) gums are considered pathological products, whereas mucilages are physiological products [47]. In addition, their similarities are related to their broad range of physicochemical properties [40,50].

Many natural gums form three-dimensional polymer networks known as gels. In this conformation, the heterogeneous gum molecules become entangled with each other (and any other large molecules also present in the structure), interfering in their movement [46,48,51]. Also, natural gums are often known for their swelling properties; in this case, properties associated with the entrapment of large amounts of water between their chains and branches. The ability to act both as solution and as gel suggests that the physicochemical characteristics of polysaccharides can be used to formulate different matrices, including membranes (films and coatings), scaffolds, gels, and hydrogels [47,51,52].

Studies involving the use of polysaccharide as membranes, gels, and hydrogels have already been developed for important and different purposes. In the pharmaceutical field, for example, polysaccharide-based membranes have been developed as wound dressings for healing treatments due to their characteristics of biomolecule immobilization, controlled release, and adhesion [52,53,54]. In addition to have healing actions *per se*, these polysaccharide formulations can act as transdermal drug delivery systems, controlling the release of active ingredients [55,56,57,58,59]. In the following sections, we discuss the use of some polysaccharide-based formulations in the healing of infected wounds using animal models and clinical trials (as summarized in Table 3).

## 3. Alginate-Based Formulations

Alginate is polymer formed by residues of β-d-mannuronate and α-l-guluronate, which is mainly extracted from brown algae (such as *Laminaria hyperborea*, *Ascophyllum nodosum,* and *Macrocystis pyrifera*) [39]. This polysaccharide is biocompatible, relatively inexpensive, and its formulations have high absorption capacities [39]. These characteristics, associated with the fact that alginate forms gels with structural similarity to the extracellular tissue matrices, make alginate an interesting material for wound dressings [75]. Following, examples of healing action of alginate-based formulations in skin wound infection are provided.

A clinical study was designed to evaluate the effects of the treatment with silver-releasing hydroalginate dressing (Silvercel) in patients with infected venous leg ulcers or pressure ulcers. The patients (*n =* 99) were divided into two groups submitted to the treatment with either Silvercel (test group; 41 subjects) or Algosteril (pure calcium alginate dressing; control group with 48 individuals) for four weeks. The authors reported that Silvercel treatment induced a faster closure rate and fewer cases of clinical infection in relation to control group. Furthermore, the wound severity score was lower in the test group [60].

Similar results were observed later in a randomized study using silver alginate powder (silver and calcium alginate powder) for the treatment of chronic wounds (leg and foot ulcers with more than one month of duration). The patients were allocated in the test group (*n =* 24); treated with foam dressing with silver alginate powder) and control group (*n =* 10); treated with foam dressing without silver alginate powder). The study was conducted for four weeks with up to three dressing changes per week. The individuals in the test group showed greater reductions in the infection score and wound surface reduction when compared to control group [61]. It is important to highlight that these above-mentioned clinical studies did not provide any data about the identification of microbial species involved in the wound infections.

Alginate-based dressings were also evaluated in experimental models of wound infections using mice and rats [32,62]. For instance, the antimicrobial action of an alginate sulfate-hydrogel containing the synthetic peptide CM11 (WKLFKKILKVL-NH_2_) was evaluated in a murine skin infection induced by a subcutaneous injection of methicillin-resistant *S. aureus* (MRSA) [32]. CM11 was chosen due to its potent antimicrobial action against several species including *A. baumannii*, *Brucella melitensis*, *E. coli*, *S. aureus*, and *P. aeruginosa* [76,77,78]. The study used ICR mice (female, eight weeks) that received a subcutaneous injection of MRSA suspension (200 μL; at 3 × 10^8^ CFU/mL) in the scapular area. The wounds were treated using a hydrogel incorporated with CM11 (128 mg/L) or hydrogel without the peptide. The treatment with CM11-loaded hydrogel improved the healing process when compared with mice without treatment. At the end of the treatment, the mice treated with peptide formulation did not show any wounds. These effects were similar to those observed for topical treatment with 2% mupirocin (positive control) [32].

In another study, alginate hydrogel was incorporated with antimicrobial honey collected in Iran for the treatment of burn infection induced by biofilm forming strains. Initially, the authors evaluated the antimicrobial action of different honeys, leading to the selection of the Thymol-based honey collected in Damavand province. The in vivo experimental model used Wistar rats (female, 6–8 weeks). Burn lesions were induced with heated steel disks (2 cm) for 9–11 s, followed by contamination with 100 µL of microbial suspension (at 1.5 × 10^5^ CFU/mL) of each bacteria (*A. baumannii*, *Klebsiella pneumoniae*, *P. aeruginosa,* or *S. aureus*). After 24 h, the mice received five grams of honey-based hydrogel (twice a day). The honey-incorporated alginate hydrogel shortened the closure period the wounds infected with all tested pathogens, when compared with the untreated mice infected with each bacteria [62].

Although the results obtained with the two above-described alginate-based hydrogels are promising, the authors did not report some important aspects related to the treatment such as bacterial loads, histological analysis, and immune response. These data are very important in order to better characterize the efficacy of the treatment.

## 4. Cellulose-Based Formulations

Cellulose is a hydrophilic polysaccharide known as the most abundant renewable organic polymer in the world. It is found in plant cell walls and in some marine organisms, bacteria, algae, fungi, and invertebrates. This carbohydrate and its derivatives have several attributes that makes them interesting as materials for wound dressing, such as bio-degradability, biocompatibility, high moisture content, high surface area, flexibility, and mechanical stability [38,79]. In addition, cellulose has several hydroxyl groups available to form hydrogen bonds that allow the chains to form ordered structures and binding of different materials to the polymeric matrix [38].

Sodium carboxymethylcellulose (SCMC) is a cellulose derivative (sodium salt of carboxymethyl ether cellulose) that has been employed for wound healing [33,80]. Films prepared from SCMC with different molecular weights were evaluated in skin infection induced in Sprague–Dawley (SD) male rats (three months). The wounds were prepared in mice back using circular plastic ring (2.20 cm × 1.27 cm) containing hot water (65.0 ± 5.0 °C). Each wound was inoculated with 40 µL of *P. aeruginosa* or *S. aureus* suspension (10^6^ CFU). The animals received the respective film 2 min after the infection, and the films were changed every 6 h for 48 h. The films did not interfere with the bacterial growth in vitro. However, all SCMC films reduced the time needed for healing the *P. aeruginosa*-infected wounds in relation to untreated mice; however, the best results were obtained by the treatment with low molecular weight SCMC (LV) films. These results were confirmed by the reduced levels of bacterial load in LV-treated mice. In relation to *S. aureus* infection, the higher activity was observed for the topical administration of high weight SCMC (HV) films [33].

Other cellulose derivative used for development of wound dressings is hydroxypropyl cellulose [63,81]. A gel based on this carbohydrate was prepared for the incorporation of the synthetic antimicrobial peptide PXL150 and tested in third-degree burn wounds infected with *P. aeruginosa* [63]. The skin of BALB/cOlaHsd mice (7–8 weeks, female) was wounded using metallic rod (1 cm^2^) heated in boiling water for 30 s. The infection was induced 5 min after wound induction by inoculation of *P. aeruginosa* PAOI-Lux1 (bioluminescent strain, 5 × 10^6^ CFU). The healing and antimicrobial effects of PXL150 were improved by its incorporation in the hydroxypropyl cellulose gel. The authors applied gels containing different concentrations of PXL150 (1.25, 2.5, 5, 10, and 20 mg/g) twice a day and observed positive results in bacterial load in all concentrations (except for 1.25 mg/g) in relation to untreated wounds [63]. PXL150 has been also showed to accelerate the healing in experimental models of surgical site infections and skin and soft tissue infections [82,83].

Cellulose in association with collagen was used for the fabrication of nanocrystals composite scaffolds containing curcumin-loaded gelatin microspheres (Cur/GMs/Coll-CNCs) [34]. This approach was employed due to the low stability and high hydrophobicity of curcumin, a compound highlighted for high therapeutic potential as antimicrobial, healing, and anti-inflammatory agent [84,85,86,87]. The authors showed that curcumin was released in a controlled and sustained way from the scaffolds. Cur/GMs/Coll-CNCs also exhibited in vitro inhibitory action towards *E. coli*, *S. aureus,* and *P. aeruginosa*. In the experimental model, full-thickness burns were induced in SD male rats using hot circular copper billets (15 mm diameter; 90 °C) during 20 s. The wounds were infected with *P. aeruginosa* suspension (1 × 10^8^ CFU). The Cur/GMs/Coll-CNCs scaffolds shortened the healing process in relation to untreated rats. The Cur/GMs/Coll-CNCs-treated animals also had lower inflammatory signals, a condition confirmed by the reduced levels of inflammatory cytokines (IL-lβ, IL-6, and TNFα) [34].

## 5. Chitosan-Based Formulations

Due to its special set of biological properties (including biocompatibility, reduced to absent toxicity, and immunostimulatory activities), chitosan has been pointed as an attractive tool for wound healing [43,88]. This polymer is obtained by partial deacetylation of chitin, the most abundant natural polymer after cellulose, as it is found in exoskeleton of certain living organisms like crustaceans and cell wall of fungi. Chitosan and its derivatives are among the most frequently natural material used for wound dressings applications [42,89]. Hybrid materials based on chitosan or its derivatives polymers have been also used for incorporate antimicrobial agents [90,91].

Several evidences have demonstrated that the healing properties of chitosan are related to its ability to modulate cellular processes related to cell proliferation and host immunity [92]. For example, this polymer promotes migration of polymorphonuclear neutrophils (PMNs), and proliferation of dermal fibroblasts [92,93]. In addition, chitosan provides additional protection towards infection due to its direct inhibitory action against several microorganism and causing minimal adverse effects [64,94]. All these attributes advocate for the use of chitosan and derivatives for the development of dressings for treatment of skin infections.

Chitosan acetate is used for preparation of bandages with high healing activity due to their hemostatic and antimicrobial actions. A comparative analysis of the healing effects of HemCon^®^ (an engineered chitosan acetate dressing), alginate sponge bandage, and silver sulfadiazine cream was performed using full-thickness excisional wounds induced in male Balb/c mice and infected with bioluminescent strains of *S. aureus* (25 × 10^7^ CFU), *P. aeruginosa* (5 × 10^7^ CFU) or *Proteus mirabilis* (25 × 10^7^ CFU). HemCon^®^ treatment was able to reduce the levels of bacterial loads in the skin. For the models using *P. aeruginosa* and *P. mirabilis*, HemCon^®^ prevented the establishment of systemic infection, resulting in the survival of all animals. This effect was not observed for the other groups, where the mice survival was reduced [64].

In a similar work (using the same experimental model described above [64]), the efficacy of HemCon^®^ for treatment of full-thickness excisional wounds infected with *S. aureus* was correlated with a reduction in the number of inflammatory cells in the wound [65]. The same research group also reported that topical administration of HemCon^®^ in BALB/c mice (female, 6 to 8 weeks) accelerated the healing of third-degree burns created by brass blocks (10 × 10 mm; preheated at 92–95 °C) and infected by *S. aureus*, *P. aeruginosa,* or *P. mirabilis*. HemCon^®^ bandage increased the survival of mice with wounds infected by *P. aeruginosa* or *P. mirabilis* in relation to those animals without treatment [66].

Another application of chitosan is the improvement of the therapeutic value of silver nanoparticles (AgNPs) [67]. Silver is an antimicrobial agent widely used for management of bacterial skin infections [95]. The authors first showed that chitosan acetate and nanoparticle silver exhibited synergistic in vitro effects against MRSA, *P. aeruginosa*, *P. mirabilis*, and *A. baumannii*. Following this, third-degree burns were induced in BALB/c mice (female; 6–8 weeks) as reported for Dai et al. [66]. *P. aeruginosa* suspension (1 × 10^8^ CFU) was inoculated in each wound five minutes after the skin damage. The treatment with chitosan acetate dressing incorporated with AgNPs accelerated the healing of *P. aeruginosa*-infected burns in relation to animals treated with chitosan acetate bandage without AgNPs or untreated mice. The use of AgNPs-chitosan acetate dressing reduced the levels of *P. aeruginosa* in blood and the mice mortality when compared with the groups treated with chitosan acetate alone [67].

It has also been reported that chitosan is a good stabilizer for AgNPs and this preparation has improved healing activity [35,96]. A mouse model of wound infection induced by a MRSA was employed to evaluate the healing efficacy of low molecular weight chitosan-coated silver nanoparticles (LMWC-AgNPs). Full-thickness skin wounds (1.5 × 1.5 cm) were created in the back of Balb/c mice followed by inoculation of 100 μL of MRSA suspension at 9 × 10^8^ CFU/mL. LMWC-AgNPs showed in vitro antimicrobial action and this preparation was less toxic towards human fibroblast than polyvinylpyrrolidone-coated silver nanoparticles (PVP-AgNPs) and uncoated AgNPs. Similarly, LMWC-AgNPs exhibited in vivo inhibitory action against MRSA and improved the healing process in relation to untreated mice. Importantly, the adverse effects typically associated with AgNPs on liver function were reduced in mice treated with LMWC-AgNPs [68].

Chitosan has been also applied for the development of hydrogel formulations loaded with therapeutics agents (such as lysostaphin and colistin) in order to manage wound infections [36,69]. Lysostaphin was incorporated into a chitosan-collagen hydrogel (CCHL) and tested against MRSA-induced burn infection in New Zealand White rabbits. The authors used an electronic temperature controller (80 °C, 15 s) to create third-degree burn wounds (3 × 3 cm). The eschar was removed two days after burning and each wound was infected with MRSA (200 µL of 1 × 10^9^ CFU/mL bacterial suspension). The animals treated with CCHL should improvements in lesion healing associated with the reduction of MRSA burden when compared to groups treated with chitosan-collagen hydrogel without lysostaphin or saline-treated animals. The administration of CCHL also leaded to a better restoration of tissue architecture at the end of the treatment [36].

Colistin was loaded into self-healable hydrogel prepared with glycol chitosan and aldehyde-modified poly(ethylene glycol) derivative and used for treatment of *P. aeruginosa*-induced burn infection in neutropenic mice. The animals treated with colistin-loaded chitosan hydrogel showed reduced levels of bacteria in the wound tissue when compared with groups treated with saline or chitosan hydrogel. In fact, colistin-loaded chitosan hydrogel promoted a similar inhibitory profile to the colistin solution for both colistin-sensitive and colistin-resistant *P. aeruginosa* strains [69].

Another application of chitosan derivatives is the loading of photosensitizer agent for photodynamic therapy (PDT). In this case, carboxymethyl chitosan nanoparticles (CMCNPs) were loaded with ammonium methylbenzene blue (MB) and used in PDT against *S. aureus* and *P. aeruginosa*. In vitro, the PDT with CMCNPs had bactericidal and biofilm eradication properties, related to ROS production. Photodynamic therapy with CMCNPs was also evaluated in vivo using

Japanese big ear rabbits submitted to subcutaneous injection of *S. aureus* (0.5 mL of 1 × 10^7^ CFU/mL).

The treatment with PDT plus CMCNPs showed high efficiency in managing bacterial infection when compared with the other experimental groups (animals treated with laser, CMCNPs or untreated). The rabbits treated with PDT plus CMCNPs also showed marked reductions on TNF-α and IL-6 levels and greater resolution of the wound in relation to other groups [70].

A nanofiber composed of poly(lactic-co-glycolic acid)-hydroxypropyltrimethyl ammonium chloride chitosan (PLGA-HACC) was also evaluated against *S. aureus*-infected wounds in mice. PLGA-HACC exhibited higher in vitro antibacterial activity (against *S. aureus* and *P. aeruginosa*) than membranes produced by PLGA alone or chitosan graft plus PLGA (PLGA-CS). PLGA-HACC also showed favorable properties such as cytocompatibility and significantly stimulated adhesion, spreading, and proliferation of fibroblasts (HDFs) and human keratinocytes (HaCaTs). In vivo studies were performed using BALB/c mice (four weeks old) in a model of full-thickness excisional wounds infected with 100 μL of *S. aureus* suspension (1 × 10^8^ CFU/mL). PLGA-HACC induced a faster contraction of *S. aureus*-infected wounds than the other tested membranes (PLGA and PLGA-CS). This effect was confirmed by the lower levels of bacteria in skin (total eradication was observed after seven days of treatment) and higher collagen deposition [71].

## 6. Hyaluronic Acid-Based Formulations

Hyaluronic acid (HA) is a non-sulfated glycosaminoglycan composed by repeated disaccharides of β-1,4-linked D-glucuronic acid and β-1,3-linked *N*-acetyl-d-glucosamine [97]. It is found as the main component of extracellular matrix in hydrated tissues, including skin where it plays essential roles in cellular migration and inflammation during tissue regeneration [97,98,99,100]. This naturally occurring polymer has some characteristics (wide availability, biocompatibility, and safety) that make it a suitable material for wound dressings [101,102,103]. The topical injection of topical hyaluronic acid in combination with cefazolin has been shown to reduce the severity of *S. aureus*-induced surgical-site infection [104].

Recently, a cross-linked hyaluronic acid-based hydrogel with EDTA−Fe^3+^ complexes incorporated with platelet derived growth factor (PDGF-BB; a mitogenic agent) was effectively employed to treat wound infection provoked by *E. coli*. The inclusion of EDTA−Fe^3+^ complexes in hydrogel aimed to promote bacterial inhibition, since during the inflammatory phase the immune cells produce hydrogen peroxide that in turn reacts with Fe^3+^ leading to production of hydroxyl radicals (through Fenton reaction). In vitro analysis showed that the hydrogel inhibited *E. coli* and *S. aureus* growth. The in vivo model used C57BL/6 mice (eight weeks old, female) and was based on the infection of full-thickness excisional wounds by *E. coli* (100 μL of a suspension at 1 × 10^9^ CFU/mL). The mice treated with the hydrogel loaded with PDGF-BB showed a faster tissue reparation than the control groups (animals treated with hydrogel without PDGF-BB or untreated mice). This effect was accomplished with improvement of angiogenesis, reduction of inflammatory burden, and bacteria levels in the skin [72].

Another hydrogel was produced by a combination of HA and dextran; this formulation was functionalized with gelatin microspheres loaded with sanguinarine (SA/GMs/Dex-HA). The alkaloid sanguinarine (purified from roots of plants as *Sanguinaria canadensis* L. and *Chelidonium majus* L.) was chosen due to its antimicrobial (related to DNA damage) and anti-inflammatory effects [105,106,107,108]. Firstly, the in vitro antimicrobial action of the formulation was evaluated against MRSA and *E. coli*, along with the mechanical properties and drug release pattern. In summary, the in vitro assay demonstrated that the hydrogel promoted a sustainable release of sanguinarine and preserved its antimicrobial activity [73].

For in vivo studies, deep partial thickness burn was induced in the back of male SD rats with a hot circular copper billet (15 mm diameter, 90 °C, 15 s). The skin was infected with MRSA suspension (100 µL; 1 × 10^8^ CFU) one day after the burn induction. The obtained results showed that SA/GMs/Dex-HA hydrogel favored the healing process of burn lesion infected by MRSA when compared with rat treated with dextran-hyaluronic acid hydrogel or dextran-hyaluronic acid hydrogel incorporated with sanguinarine. In fact, the mice treated by SA/GMs/Dex-HA showed better profiles of re-epithelialization and extracellular matrix remodeling [73]. Furthermore, the treatment down-regulated the amounts of TGF-β1 (transforming growth factor-β1; related with hypertrophic scars and keloids formation due to myofibroblast differentiation [109,110]) and TNF-α; and up-regulated the expression of TGF-β3 (transforming growth factor-β3; inductor of collagen synthesis and fibroblast proliferation [111]). The authors did not report data about bacterial load in the tissue [73].

Based on all positive results obtained from in vitro and in vivo model, a recent clinical trial evaluated the efficacy of a topical spray containing HA and metallic silver (Hyalosilver, Fidia Farmaceutici S.p.A, Abano Terme, Italy) in human lesion (vascular or pressure ulcers) with signs of bacterial colonization. The treatment was based on the daily self-application of the spray for 28 days. The evaluations of wound size, bacterial load, and possible adverse effects were performed at days 1, 7, and 28. The authors showed that the spray application (once a day for 28 days) efficiently reduced the area of chronic wounds and bacteria contamination. The patients treated with the spray also had better clinical parameters (odor, exudate, and erythema) [74].

## 7. Conclusions

The data analyzed in this manuscript clearly point out that polysaccharides (such as alginate, cellulose, chitosan, hyaluronic acid) are versatile polymers for the development of formulations to treat skin infections. The efficacy of these biomolecules is related to their properties such as biocompatibility, biodegradability, poly-functionality, high chemical reactivity, chirality, chelation, and adsorption capacity. Among the carbohydrates, chitosan (and its derivatives) was the most employed in in vivo studies. In most of the cases, the high therapeutic indexes obtained may be results of synergistic action of the polymer and the incorporated agent (antimicrobial, healing, or immunomodulatory compounds). In summary, these polysaccharide-based formulations can play a role in the first line treatment for infected wounds, or act as adjuvant therapy in combination to traditional methods.

## Figures and Tables

**Figure 1 biomolecules-10-00063-f001:**
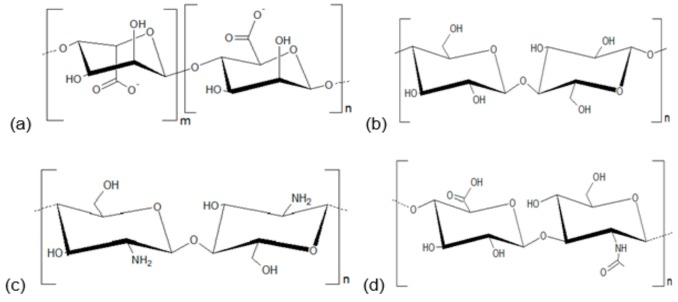
Structural representations of the polysaccharides used for development of dressings evaluated in this study. (**a**) Alginate; (**b**) cellulose; (**c**) chitosan; (**d**) hyaluronic acid.

**Table 1 biomolecules-10-00063-t001:** Summary of fundamental and secondary characteristics for an ideal dressing.

Fundamental Characteristics	Secondary Characteristics
1- It should offer mechanical protection; 2- It should keep the wound with optimum moisture and pH; 3- It should allow gas exchange with the environment; 4- It should ensure biological barrier; 5- It should have affordable cost; 6- It should be water resistant and easily removable; 7- It should not need frequent changes; 8- It should not be toxic or cause allergic reactions; 9- It should not release non-biodegradable particles or fibers into the wound. 10- It should not cause pain when applied or removed.	1- It could promote wound healing; 2- It could have antimicrobial activity; 3- It could have hemostatic activity; 4- It could promote the debridement; 5- It should minimize the scar tissue formation; 6- It could be transparent to allow healing monitoring; 7- It could inactivate proteolytic enzymes in chronic wounds; 8- It could be able to absorb bad odor.

Based on [13,14,15,16,17,18,19].

**Table 2 biomolecules-10-00063-t002:** Advantages and disadvantages of some dressings used for the treatment of wounds.

**Formulation Type**	**Indications**	**Advantages**	**Disadvantages**	**References**
**Foams**	They are indicated for wounds with moderate to high levels of exudates.	They are semipermeable and have good porosity; They provide thermal insulation; They ensure a moist environment; They are suitable for sensitive and fragile skin; They have good absorption capacity; They can reduce over granulation; They allow atraumatic removal.	They do not offer mechanical protection; They are not suitable for treatment of burns.	[13,19,25,26,27]
**Hydrogels**	They are indicated for dry necrotic wounds and minimally exuding wounds.	They have high water content and maintain a moist environment; They are clear or transparent, allowing the wound monitoring; They promote re-epithelialization; They facilitate autolytic debridement of necrotic area.	They have weak mechanical properties; They are used in combination with a secondary dressing (such as foams).	[19,27,28,29]
**Hydrocolloids**	They are suitable for partial- or full-thickness acute and chronic wounds.	They maintain a moist environment; They absorb minimal to moderate amounts of drainage; They are easy to be removed; They facilitate autolytic debridement of necrotic area; They contribute to pain management; They provide a barrier to external microorganism; They promote re-epithelialization; They promote acidification which can inhibit bacteria growth.	They can have toxicity; They have weak mechanical properties; They can have unpleasant odor.	[19,27,30]
**Films**	They can be used as primary or secondary dressings. As primary dressing, they are indicated for dry, superficial wound. As a secondary dressing they can be used combined with foam dressings in heavier exuding wounds.	They are semipermeable (impermeability to water, bacteria, and dirt; but permeable to wet vapors); They maintain a moist environment; They are flexible; They are clear or transparent, allowing the wound monitoring.	They do not absorb exudates; They generally require a border of dry, intact skin for application (for adhesiveness); They can damage the epidermal skin layer during the removal.	[19,20,27,31]
**Membranes**	They are indicated for infected wounds with moderate to heavily exuding wounds.	They have high capacity to absorb exudate (up to 20 times their weight); They are permeable to wet vapors; They maintain a moist environment; They are flexible and biodegradable; They minimize bacterial infection.	They can cause excessive dehydration (not indicated to dry wounds); Some membranes require secondary dressings (to avoid dryness).	[19,27,31]

**Table 3 biomolecules-10-00063-t003:** Polysaccharide-based formulations employed in the treatment of skin-related infections in animal models and clinical trials.

Polysaccharide	Co-Polymer	Formulation Type	Incorporated Agent	Infection Model	Bacteria	Ref.
Alginate	-	Alginate dressing	Silver	Human	-	[60]
		Alginate dressing	Silver	Human	-	[61]
	-	Alginate dressing	CM11 peptide	Animal (mice)	MRSA	[32]
	-	Hydrogel	Honey	Animal (rats)	*A. baumannii*; *K. pneumoniae*; *P. aeruginosa*; *S. aureus*	[62]
Cellulose (Sodium carboxymethyl cellulose)	-	Film	-	Animal (rats)	*P. aeruginosa*; *S. aureus*	[33]
Cellulose (Hydroxypropyl cellulose)	-	Gel	PXL150 peptide	Animal (mice)	*P. aeruginosa*	[63]
Cellulose	Collagen	Scaffolds	Curcumin (loaded in gelatin microspheres)	Animal (rats)	*E. coli*; *P. aeruginosa*; *S. aureus*	[34]
Chitosan acetate	-	Dressing	-	Animal (rats)	*P. mirabilis; P. aeruginosa*; *S. aureus*	[64,65,66]
	-	Dressing	Silver nanoparticles	Animal (mice)	*A. baumannii*; MRSA; *P. mirabilis; P. aeruginosa*	[67]
Chitosan		Dressing	Silver nanoparticles	Animal (mice)	MRSA	[68]
	Collagen	Hydrogel	Lysostaphin	Animal (rabbits)	MRSA	[36,69]
Chitosan (glycol chitosan)	Aldehyde-modified poly(ethylene glycol) derivative	Hydrogel	Colistin	Animal (mice)	*P. aeruginosa*	[36,69]
Chitosan (Carboxymethyl chitosan)	-	- Nanoparticles	-	Animal (rabbits)	*P. aeruginosa*; *S. aureus*	[70]
Chitosan (Hydroxypropyltrimethyl ammonium chloride chitosan)	Poly(lactic-co-glycolic acid)	Nanofibrous Membranes	-	Animal (mice)	*S. aureus*	[71]
Hyaluronic acid	-	Hydrogel	EDTA−Fe3+; PDGF-BB growth factor	Animal (mice)	*E. coli*; *S. aureus*	[72]
	Dextran	Hydrogel	Sanguinarine (loaded in gelatin microspheres)	Animal (mice)	*E. coli*; MRSA	[73]
	-	Topical spray	Metallic silver	Human	-	[74]

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
