# Peer review of "Polysaccharide-Based Formulations for Healing of Skin-Related Wound Infections: Lessons from Animal Models and Clinical Trials"

_biomolecules, 2019, doi:10.3390/biom10010063_

Round 1

Reviewer 1 Report

Scientific issues:

line 55: how was the search performed, what were the search criteria table: what were the effects that were found in the studies line 287: There has hardly been any evidence described, showing that traditional methods could be replaced by polysaccharide-based formulations. At least one study shows even a less efficient healing (ref 14). It would be more appropriate to state that polysaccharide-based formulations could play a role in the first line treatment, or as adjuvant therapy next to traditional methods.

Grammatical adjustments to be made:

line 26: controlling (instead of control) line 32: occur (instead of occurs) line 34: environment (instead of environmental) line 36: leading to the development of (chronic) wounds line 36: contribute to (instead of contribute with) line 41: Chronic (instead of The chronic) line 44: the biofilm is a favorable line 47: to combat (instead of for combat) line 70: is (instead of are) line 70: three properties?; but 4 are given in the following sentences. line 104: that (instead of of) line 105: forms gels (or forms a gel), (instead of form gel) line 113: furthermore (instead of further) lines 116 and 130: for the treatment (instead of for treatment) line 131: leading to (instead of leading for) line 134: period of wounds infected with (instead of period the wounds infected will) line 140: is a hydrophilic (instead of is hydrophilic)  line 143: them interesting as materials (instead of them as interesting materials) line 168: allowed for (a) controlled (instead of allowed the controlled) line 182: interact (instead of interacts) line 186: for the development (instead of for development) lines 199 and 259: Another (instead of Other) line 200: for the development (instead of for development) lines 206 and 215: also been (instead of been also) line 208: to evaluate (instead of for evaluate) line 211: toxic (instead of toxicity) line 220: at the end (instead of in the end) line 223: profile to (instead of profile than) line 228: related to (instead of related with) line 229: rabbits (instead of rabbit) line 230: managing (instead of manage) line 246: occurring (instead of occurred) line 247: a suitable material for (instead of as suitable material of) line 247: topical injection of hyaluronic (instead of topical injection of topical hyaluronic) line 248: has been shown (instead of was shown) line 251: effectively (instead of effective) line 259: by a combination (instead of by combination) line 279: point out (instead of pointed) lines 124, 166, 185, 188, 261: due to (instead of due)

Author Response

Dear reviewer,

Thank you so much for your contributions. We provide bolow  a point-by-point response for your questions.

Best regards.

Reviewer 1

line 55: how was the search performed, what were the search criteria table: what were the effects that were found in the studies

line 287: There has hardly been any evidence described, showing that traditional methods could be replaced by polysaccharide-based formulations. At least one study shows even a less efficient healing (ref 14). It would be more appropriate to state that polysaccharide-based formulations.

Our response: We have changed the conclusion following your suggestion.

Grammatical adjustments to be made:

line 26: controlling (instead of control) line 32: occur (instead of occurs) line 34: environment (instead of environmental) line 36: leading to the development of (chronic) wounds line 36: contribute to (instead of contribute with) line 41: Chronic (instead of The chronic) line 44: the biofilm is a favorable line 47: to combat (instead of for combat) line 70: is (instead of are) line 70: three properties?; but 4 are given in the following sentences. line 104: that (instead of of) line 105: forms gels (or forms a gel), (instead of form gel) line 113: furthermore (instead of further) lines 116 and 130: for the treatment (instead of for treatment) line 131: leading to (instead of leading for) line 134: period of wounds infected with (instead of period the wounds infected will) line 140: is a hydrophilic (instead of is hydrophilic)  line 143: them interesting as materials (instead of them as interesting materials) line 168: allowed for (a) controlled (instead of allowed the controlled) line 182: interact (instead of interacts) line 186: for the development (instead of for development) lines 199 and 259: Another (instead of Other) line 200: for the development (instead of for development) lines 206 and 215: also been (instead of been also) line 208: to evaluate (instead of for evaluate) line 211: toxic (instead of toxicity) line 220: at the end (instead of in the end) line 223: profile to (instead of profile than) line 228: related to (instead of related with) line 229: rabbits (instead of rabbit) line 230: managing (instead of manage) line 246: occurring (instead of occurred) line 247: a suitable material for (instead of as suitable material of) line 247: topical injection of hyaluronic (instead of topical injection of topical hyaluronic) line 248: has been shown (instead of was shown) line 251: effectively (instead of effective) line 259: by a combination (instead of by combination) line 279: point out (instead of pointed) lines 124, 166, 185, 188, 261: due to (instead of due)

Our response: All these adjustments were made.

Reviewer 2 Report

This is a well written review of the literature about the use of polysaccharide- based formulations for wound healing to avoid skin infections. Selected clinical trials and animal models testing alginate, (hydroxypropyl and carboxymethyl) cellulose, chitosan, hyaluronic acid are reviewed. It might be interesting for people working in this field.

Author Response

Dear reviewer,

Thank you so much for your comments.

Best regards.

Reviewer 3 Report

In the present manuscript the authors review some in vitro and/or in vivo studies using polysaccharide-based formulations for the treatment of skin wounds infections that are listed in Table 1.

Experimental details of in vitro and in vivo studies, described are missing.  

This is just an example : “The in vivo experimental model was based on the contamination (with A. baumannii, Klebsiella pneumoniae, P. aeruginosa or S. aureus) of burn lesions in rats.” How were the burn lesions induced ?“The incorporated alginate hydrogel shortened the closure period the wounds infected will all tested pathogens [45]. “ In comparison with what ?

A strong revision, adding information missing, in all document should be performed including more experimental details.

A small section should be included describing what are the therapies approved for this pathology and which are the main drawbacks associated to the ones in clinical use.

Moreover, a strong English revision should be performed in all document.

Author Response

Dear reviewer,

Thank you so much for your contributions. We provide bolow  a point-by-point response for your questions.

Best regards.

Reviewer 3

In the present manuscript the authors review some in vitro and/or in vivo studies using polysaccharide-based formulations for the treatment of skin wounds infections that are listed in Table 1.

Experimental details of in vitro and in vivo studies, described are missing. 

This is just an example : “The in vivo experimental model was based on the contamination (with A. baumannii, Klebsiella pneumoniae, P. aeruginosa or S. aureus) of burn lesions in rats.” How were the burn lesions induced ?“The incorporated alginate hydrogel shortened the closure period the wounds infected will all tested pathogens [45]. “ In comparison with what ?

A strong revision, adding information missing, in all document should be performed including more experimental details.

Our response: We have included more details for each work, please see the highlighted sentences in the updated manuscript.

A small section should be included describing what are the therapies approved for this pathology and which are the main drawbacks associated to the ones in clinical use.

Our response: We have included a discussion about the dressings used in the therapy (please see Lines 48-53; 57-68; and tables 1 and 2).

Moreover, a strong English revision should be performed in all document.

Our response: We have performed a revision in the text.

Reviewer 4 Report

The manuscript entitled “Carbohydrate-based formulations for healing of skin related wound infections: lessons from animal models and clinical trials” could be regarded as mini review type manuscript concerning the emerging field of skin related wound infections treatment by polysaccharide formulations and hybrid materials based on. The review is relatively well structured though my first remarks is the recommendation of starting with cellulose based materials review as the cellulose is the most common and abundant polysaccharide on the Earth. Another critical remark includes:

1. The suggest title of the review is somehow incorrect as usually as carbohydrates are denoted the low-molecular compounds, thus I would suggest the use of term “carbohydrate polymer-based formulations “instead.

2. In the chitosan part of the review, I would recommend including the following papers, concerning research on hybrid antibacterial nanofibrous materials based on chitosan / N-carboxyethyl chitosan and silver nanoparticles, in situ cross-linked with glutaraldehyde:  Electrospun Hybrid Nanofibers Based on Chitosan or N‐Carboxyethylchitosan and Silver Nanoparticles; Macromolecular bioscience 9. 9. 884-894 (2009); Hybrid nanofibrous yarns based on N-carboxyethylchitosan and silver nanoparticles with antibacterial activity prepared by self-bundling electrospinning; Carbohydrate research 345, 16, 2374-2380 (2010).

Author Response

Dear reviewer,

Here we provide a point-by-point responses for your questions.

Thank you so much for your contributions. 

Reviewer 4

The manuscript entitled “Carbohydrate-based formulations for healing of skin related wound infections: lessons from animal models and clinical trials” could be regarded as mini review type manuscript concerning the emerging field of skin related wound infections treatment by polysaccharide formulations and hybrid materials based on. The review is relatively well structured though my first remarks is the recommendation of starting with cellulose based materials review as the cellulose is the most common and abundant polysaccharide on the Earth.

Our response: We choose to organize the polysaccharide by alphabetic order, with the purpose to make easier to the reader to find the polymer information.

Another critical remark includes:

The suggest title of the review is somehow incorrect as usually as carbohydrates are denoted the low-molecular compounds, thus I would suggest the use of term “carbohydrate polymer-based formulations “instead.

Our response: We have changed the title for “Polysaccharides-based formulations for healing of skin-related wound infections: lessons from animal models and clinical trials”

In the chitosan part of the review, I would recommend including the following papers, concerning research on hybrid antibacterial nanofibrous materials based on chitosan / N-carboxyethyl chitosan and silver nanoparticles, in situ cross-linked with glutaraldehyde: Electrospun Hybrid Nanofibers Based on Chitosan or N‐Carboxyethylchitosan and Silver Nanoparticles; Macromolecular bioscience 9. 9. 884-894 (2009); Hybrid nanofibrous yarns based on N-carboxyethylchitosan and silver nanoparticles with antibacterial activity prepared by self-bundling electrospinning; Carbohydrate research 345, 16, 2374-2380 (2010).

Our response: We have included the information of hybrid material based on chitosan.

Round 2

Reviewer 3 Report

A strong revision on the manuscript was performed improving the quality of the review.

I would suggest that in Table 1 (revised manuscript)  the verbs "it should"  and "it could" should be removed. However, I will leave this suggestion to the Editor.